# Detecting Trivariate Associations in High-Dimensional Datasets

**DOI:** 10.3390/s22072806

**Published:** 2022-04-06

**Authors:** Chuanlu Liu, Shuliang Wang, Hanning Yuan, Yingxu Dang, Xiaojia Liu

**Affiliations:** 1School of Computer Science and Technology, Beijing Institute of Technology, Beijing 100081, China; 3120160469@bit.edu.cn (C.L.); yhn6@bit.edu.cn (H.Y.); 3220200865@bit.edu.cn (Y.D.); lxj@bit.edu.cn (X.L.); 2Institute of E-Government, Beijing Institute of Technology, Beijing 100081, China

**Keywords:** quadratic optimized trivariate information coefficient (QOTIC), trivariate associations, maximal information coefficient (MIC), correlation, large data

## Abstract

Detecting correlations in high-dimensional datasets plays an important role in data mining and knowledge discovery. While recent works achieve promising results, detecting multivariable correlations especially trivariate associations still remains a challenge. For example, maximal information coefficient (MIC) introduces generality and equitability to detect bivariate correlations but fails to detect multivariable correlation. To solve the problem mentioned above, we proposed quadratic optimized trivariate information coefficient (QOTIC). Specifically, QOTIC equitably measures dependence among three variables. Our contributions are three-fold: (1) we present a novel quadratic optimization procedure to approach the correlation with high accuracy; (2) QOTIC exceeds existing methods in generality and equitability as QOTIC has general test functions and is applicable in detecting multivariable correlation in datasets of various sample sizes and noise levels; (3) QOTIC achieved both higher accuracy and higher time-efficiency than previous methods. Extensive experiments demonstrate the excellent performance of QOTIC.

## 1. Introduction

In the era of rapid development of information technology, information is being stored and interacted in a digital way through the continuous creation, storage and accumulation of massive data [1], while in the real world, they are often high-dimensional, noisy, and valuable associations are hidden. In particular, it is difficult to mine trivariate relationships. Such data are increasingly common in fields such as genomics, physics, and economics, making this problem an important and growing challenge [2]. To extract the associations from high-dimensional data, it is basal to identify whether the correlation among variables is strong or not. Variables with strong correlation to others are preserved for further use, and those with weak correlation are filtered out. To ensure that the important associations among variables are not missed, the relationships with different strengths are assigned to different correlations by statistical measures [3,4,5,6,7,8,9,10,11,12,13,14,15]. The statistic to measure dependence should have the heuristic properties of generality and equitability for searching pairs of variables that are closely associated. And the equitability is more practical.

There are some methods to measure multivariable dependence, such as distance correlation (Dcor) [13,14], nonlinear correlation information entropy (NCIE) [15], maximal information entropy (MIE) [16] and maximal three-dimensional information coefficient (MTDIC) [17]. However, most of them are not designed for the equitability. Maximal information coefficient (MIC) [1] is a measure of dependence for bivariate relationships, and it gives the relationships under the same noise with similar scores. For MIC, a grid can be drawn on the scatterplot of two variables to encapsulate the relationship. When the partitioned of grids are determined, the correlation can be calculated by mutual information. It is a bivariate correlation as pairwise variables in a two-dimensional space, which may be not applicable to a trivariate correlation as three variables in a three-dimensional space. To further design a measure of dependence for three variables with equitability, there are two major challenges. One is how to partition the grids in a three-dimensional space to encapsulate the relationship, the other is how to reduce the computing time while the number of variable increases.

In this paper, quadratic optimized trivariate information coefficient (QOTIC) is proposed to measure trivariate dependence. By uncovering the effects of partitioning generality and equitability in each dimension, a quadratic optimization procedure is put forward, in which one-dimensional adaptive equal partition is combined with another two-dimensional dynamic partition to improve the equitability. QOTIC is consistent with exhaustive search trivariate information coefficient (ESTIC) with the highest accuracy, and the time loss is reduced by 28%. Compared with Dcor, MIE, NCIE, and MTDIC on the generality and equitability, QOTIC is also applied in a dataset from Global Health Observatory for exploring the factors on life expectancy.

Our contributions are three-fold: (1) we present a novel quadratic optimization procedure to approach the correlation with high accuracy; (2) QOTIC exceeds existing methods in generality and equitability as QOTIC has general test functions and is applicable in detecting multivariable correlation in datasets of various sample sizes and noise levels; and (3) QOTIC achieves balance both between accuracy and time-efficiency and outperforms previous methods. The rest of the paper is organized as follows. After this introduction, Section 2 summarizes related works in this field. Section 3 presents the definitions, properties and algorithms offered by QOTIC. Section 4 compares QOTIC with other existing methods in terms of equitability and generality. In Section 5, QOTIC is applied for the real-world dataset to show its performance. Finally, in Section 6, the conclusions of our research are drawn.

## 2. Related Works

To identify the arbitrary correlations in high-dimensional data sets, various methods are applied to find out the relationships between pairwise variables, which include MIC [1], Pearson’s correlation coefficient [3], principal curve-based method [4], maximal correlation [5], distance correlation [13,14] and so on. All of these methods have the problem of variable extension, that is, how to generalize from binary variables to multivariate variables. Introducing the generality and equitability, MIC explores the effects of an approximation algorithm and normalization on the equitability. The deviations from the original MIC values’ equitability are resulted from the approximation algorithm’s accuracy rather than the nature of MIC [6]. To improve the equitability in an acceptable run-time, Wang et al. presented an iMIC for optimizing the partition [7]. However, with the increase of sample size, the time cost is still relatively high. As a property for the measure of dependence, the equitability was formally defined and characterized by Reshef et al. [8]. Furthermore, MIC aroused more questions. Simon and Tibshirani [9] pointed out that MIC would cause false positives in data analysis due to low power. Kinney and Atwal [10] provided a mathematical proof to support mutual information as an alternative to use MIC. Focusing on the power and equitability, Reshef et al. gave MICe [11] and total information coefficient (TICe) [12]. The combination of the two statistics yields an efficient approach for obtaining both power and equitability. In 2020, equitability was characterized again in terms of interval estimates [8]. Since MIC may overestimate the correlated value, which leads to the misidentification of the relationship without noiseless, to detect unbiased associations, Liu proposed unbiased correlation measure weighted information coefficient mean (WICM) [18]. To quantify the dependence between two random vectors of possibly different dimensions, Mordant proposed two coefficients that are based on the Wasserstein distance between the actual distribution and a reference distribution with independent components, but they were not designed with the goal of equitability [19]. For the latest practical applications, Liu et al. proposed a novel method to discover the association of algae with physicochemical variables related to the algae growth in Erhai Lake, by integrating MIC and association rules [20]. Guo et al. proposed EpiMIC for epistasis detection [21].

The research on multivariable dependence is relatively few and can be categorized into three types. The first one is based on the correlation matrix. An element in the matrix is the correlation value between two variables. This is carried out by calculating the eigenvalues of the correlation matrix and then getting the overall correlation. Typical methods are the NCIE [15] on the basis of nonlinear correlation coefficients (NCC), and the MIE [16] under MIC. The extension of NCIE and MIE to multivariate variables is still based on the calculation of binary variables without actually calculating multivariate variables as a whole. The second one is the extension of bivariate measures, such as Dcor based on Euclidean distances between sample elements [13,14], and conditional multiinformation (CMI) for detecting the conditional dependence between multiple discrete variables [22]. Because they were not designed with equitability and generality as its goal, they performed poorly in these two areas. The third one does not directly give a correlation value by a statistic, but analyzes the dependency among many variables for a specific application, such as the principal component analysis [23], factor analysis [24], canonical correlation analysis [25,26], and feature selection [27].

## 3. Proposed Method

In this section, the quadratic optimized trivariate information coefficient (QOTIC) will be presented.

### 3.1. Trivariate Mutual Information

Assume that there are three random variables *X*, *Y* and *Z*. *I*(*X*; *Y*; *Z*) is their trivariate mutual information.
*I*(*X*; *Y*; *Z*) = *H*(*X*) + *H*(*Y*) + *H*(*Z*) − *H*(*X*, *Y*) − *H*(*X*, *Z*) − *H*(*Y*, *Z*) + *H*(*X*, *Y*, *Z*)(1)
where *H*(*) denotes Shannon entropy. The greater the uncertainty, the greater the entropy is. For binary distribution, the joint distribution’s uncertainty is greater than or equal to the marginal distribution, i.e., *H*(*Y*) ≤ *H*(*X*, *Y*), *H*(*Z*) ≤ *H*(*Y*, *Z*), and *H*(*X*) ≤ *H*(*X*, *Z*). In Equation (1), if and only if *H*(*X*, *Y*) − *H*(*Y*) = 0, *H*(*Y*, *Z*) − *H*(*Z*) = 0, and *H*(*X*, *Z*) − *H*(*X*) = 0, *I*(*X*; *Y*; *Z*) reaches its maximum, i.e., *H*(*X*, *Y*, *Z*).

Figure 1 shows the advantages of trivariate associations analysis. Take the analysis of the factors associated with target T among *m* factors as an example. Bivariate associations analysis measures the binary correlation between any factor and target T, and retains the factors with strong correlation with the target. Trivariate associations analysis measures the correlation between any two factors and target T. If the correlation value is high, two factors are retained. For any two factors v*_x_* and v*_y_*, if v*_x_*, v*_y_* and T are strongly correlated, but one of v*_x_* and v*_y_* is weakly correlated with T, only bivariate associations analysis will discard factor v*_x_* because the measure of correlation between v*_x_* and T is small.

### 3.2. Trivariate Characteristic Matrix

For a finite set *D*⊂*R*^3^, if the first dimension, the second dimension, and the third dimension is taken as *x*-axis, *y*-axis, and *z*-axis, respectively, cubic space comes into being. When the data in *D* is partitioned, the resulting cubic grid is (*x*, *y*, *z*), where *x*, *y* and *z* are all positive integers greater than 1. Figure 2 illustrates the strategy of partition. 

In Figure 2, QOTIC trivariate partition in cubic space is distinguished from MIC bivariate partition in plane space, and MTDIC trivariate partition in cubic space. For bivariate variables in plane space, the partition strategy of MIC is that when one axis is dynamically partitioning, keep the other axis equipartition. First, equipartition *y*-axis, and then dynamically partitioning *x*-axis. Second, equipartition *x*-axis, dynamically partitioning *y*-axis. For trivariate variables in cubic space, the partition strategy of MTDIC is that only one axis is dynamically partitioned and the other two axes keep equipartition. Equipartition *x*-axis and *y*-axis, and then dynamically partitioning *z*-axis. For trivariate variables in cubic space, the partition strategy of QOTIC is that only one axis is equipartition, and for the other two dynamic partition axes, one of them is based on the other dynamic partition. The first and the second steps of QOTIC are the same to MTDIC. When dynamically partitioning *z*-axis is finished, fix the current partition of *z*-axis, and dynamically partition *y*-axis.

Let *D*|(*x*, *y*, *z*) denote the distribution induced by points in *D* points on the cubic grid (*x*, *y*, *z*). The probability mass in a grid is the fraction of points in *D* falling into the grid. Then Equation (1) may become Equation (2):(2)$$I\left(D|\left(x,y,z\right)\right)=\overset{x}{\underset{i=1}{\sum }}\overset{y}{\underset{j=1}{\sum }}\overset{z}{\underset{k=1}{\sum }}p\left(x_{i},y_{j},z_{k}\right)log\frac{p\left(x_{i},y_{j}\right)p\left(x_{i},z_{k}\right)p\left(y_{j},z_{k}\right)}{p\left(x_{i}\right)p\left(y_{j}\right)p\left(z_{k}\right)p\left(x_{i},y_{j},z_{k}\right)}.$$
where *p*(*x*, *y*, *z*) is the fraction of *D*’s points that fall into the grid (*x*, *y*, *z*). *p*(*x*, *y*), *p*(*x*, *z*), and *p*(*y*, *z*) are, respectively, the marginal distribution of *xy*-plane (*x*, *y*, *), *xz*-plane (*x*, *, *z*), and *yz*-plane (*, *y*, *z*). *p*(*x*), *p*(*y*), and *p*(*z*) are separately the marginal distribution of *x*-axis, *y*-axis, and *z*-axis.

**Definition** **1.**
*Maximal trivariate mutual information*
*For a finite set D*⊂*R*^3^*and positive integers x, y, z:**I**(*D*|(*x*, *y*, *z*)) = max *I*(*D*|(*x*, *y*, *z*)).(3)*where I(D|(x, y, z)) denotes the trivariate mutual information under the grid (x, y, z), and its maximum I*(D|(x, y, z)) is over all possible cubic grids with x-bins, y-bins, and z-bins in the first, the second, and the third dimension, respectively*.

From Definition 1, to get the maximal trivariate mutual information can be taken as the process of finding a reasonable cubic grid. When dealing with the bivariate data, it is unfeasible to test infinite partitions, not to mention trivariate data. For MTDIC [17], the specific process of dynamic partition is implemented in two steps. The first step is to equally partition the values in *x*-axis and *y*-axis into sequence *R* and sequence *Q,* respectively. And the second step is to fix the partition on these two axes and then partition the values in *z*-axis into sequence *P* by using the iterative optimization [17]. Moreover, QOTIC’s solution to this problem is to equally partition the values in *x*-axis and *y*-axis and then optimize them on *z*-axis. After getting the optimal partition, fix the partition on *z*-axis, and conduct quadratic optimization on *y*-axis. In a word, MTDIC equipartition *x*-axis, *y*-axis and dynamically partition *z*-axis, which is a single optimization. Besides equipartitioning *x*-axis, *y*-axis and dynamically partitioning *z*-axis, QOTIC further dynamically partition *y*-axis (QuadraticApproxMI) under the partitioned *z*-axis, which is the quadratic optimization.

In the light of Definition 1, we may further get the definition of trivariate equipartition mutual information, trivariate equicharacteristic matrix, and trivariate characteristic matrix.

**Definition** **2.**
*Trivariate equipartition mutual information*

*For a finite set D*
*⊂R^3^ and positive integers x, y, z,*
*I^E^*(*D*|(*x*, *y*, *z*)) = *I*(*D*|(*x*, *y*, *z*)*_E_*).(4)
*where I^E^(D|(x, y, z)) denotes the trivariate mutual information under the cubic grid (x, y, z)_E_ that equipartitions the first, second and third dimension with x-bins, y-bins and z-bins.*
*The grid (x, y, z)_E_ is a special case of all possible grids in the search for I*(D|(x, y, z)).*


**Definition** **3.**
*Trivariate equicharacteristic matrix*

*The trivariate equicharacteristic matrix M^E^(D) of a finite set D*
*⊂R^3^ is*

(5)
$$M^{E}{\left(D\right)}_{x,y,z}=\frac{I^{E}(D|\left(x,y,z\right))}{\log\min\left\{x,y,z\right\}}\;$$



**Definition** **4.**
*Trivariate characteristic matrix*

*The trivariate characteristic matrix M(D) of a finite set D*
*⊂R^3^ is*

(6)
$$M{\left(D\right)}_{x,y,z}=\frac{I^{*}(D|\left(x,y,z\right))}{\log\min\left\{x,y,z\right\}}\;$$



The process of equipartition is called adaptive equipartition. Figure 3 shows the equitability performance of TEIC (adaptive equipartition), MTDIC (single optimization) and QOTIC (quadratic optimization) on different noises in 12 functional relationships reproduced from MTDIC [17], and Table 1 interprets the color on a relationship and the variable on an axis. Figure 4 shows comparison of bias and variance of QOTIC, MTDIC, TEIC, and Table 2 shows the analysis of bias and variance of QOTIC, MTDIC, and TEIC. For example, given an arbitrary random variable *r*, three variables on each axis are *X* = *f*_x_(*r*), *Y* = *f*_y_(*r*), and *Z* = *f*_z_(*r*), which refer to the relationship types in each axis, respectively. Each functional relationship is uniformly distributed and contains additive Gaussian noise. The noise level increases gradually, and the coefficient of determination *R*^2^ ranges from 0 to 1.

In Figure 3, the sample size of each functional relationship is 500. The equitability performance of the correlation values of the 12 functional relationships is calculated by using adaptive equipartition, single optimization, and quadratic optimization, the results of which are the left, the middle, and the right pictures. Comparing Figure 3a–c, there are three findings.

Firstly, when there is no dependency between the variables, the quadratic optimization does not improve the adaptive division and the single optimization gives a correlation value close to 0.Secondly, when there is a strong correlation between variables without noise, the quadratic optimization gives very high correlation values for all relation types, and the correlation values given to different relations are close to 1, better than the single optimization.Thirdly, when there is a dependency relationship between variables but accompanied by noise, under the same noise level, the approximate of the correlation value given by the quadratic optimization for different relationships is higher than that of the single optimization. Therefore, the quadratic optimization further encapsulates the relationship on the basis of the single optimization, and improves the performance in terms of equitability.

### 3.3. Generating the Trivariate Characteristic Matrix

Each entry corresponds to the normalized trivariate mutual information obtained by using dynamic optimization under its partition in the trivariate characteristic matrix. To generate the trivariate characteristic matrix under the given upper bound *B*(*n*) of the partition for cubic grids, the values of the *x*-axis and *y*-axis are equally partitioned first, and dynamic optimization is implemented on the *z*-axis. When getting the optimal partition on the *z*-axis, fix the current partition, and conduct QuadraticApproxMI on the *y*-axis. If the results of quadratic optimization on the *y*-axis is better than that of the equipartition on the *y*-axis under the current partition, it is retained as an entry in the characteristic matrix. Algorithm 1 shows the pseudocode to generate the trivariate characteristic matrix by using quadratic optimization.


**Algorithm 1:** Generating trivariate characteristic matrix.**Input:***D*, Parameter *c* controls the granularity of the partition**Output:**
*M*(*D*)**for** *x* = 2 to [B(n)4] **do**  Getting equipartition *R* with size *x*  **for** *y* = 2 to [B(n)2∗x] **do**  Getting equipartition *Q* with size *y*  *z =* [B(n)x∗y]  [*I*_(*x*,_ *_y_*_, 2),_ *I*_(*x*,_ *_y_*_, 3)…,_
*I*_(*x*,_ *_y_*_,_ *_z_*_)_] = Dynamic optimizing(*D*, *Q*, *R*, *z*)**for** *k* = 2 to *z* **do***M*(*D*)*_x_*_,_ *_y_*_,_ *_k_*= *I*_(*x*,_ *_y_*_,_ *_k_*_)_/log min{*x*, *y*, *k*}**end for***P_l_* = argmax{*M*(*D*)*_x_*_,*y*,*l*_|*P_l_*,2 ≤ *l* ≤ *z*}*I^T^*^′^
_(*x*,_ *_y_*_,_ *_l_*_)_ = QuadraticApproxMI(*D*,*R*,*P_l_*,*cl*)*M*(*D*)*_x_*_,_ *_y_*_,_ *_l_*= max{*I*′_(*x*, *y*, *l*)_, *I*_(*x*, *y*, *l*)_}/log min{*x*, *y*, *l*}**end for****end for**


**Definition** **5.**
*Quadratic optimized trivariate information coefficient*

*For a data set D of three variables with sample size n, quadratic optimized trivariate information coefficient of D is given by*

(7)
$$\;\mathrm{QOTIC}\left(D\right){=}_{xyz\leq B\left(n\right)}^{\;\;max}\left\{M{\left(D\right)}_{x,y,z}\right\}$$



The flow chart of QOTIC method is shown in Figure 5.

To improve the accuracy of the approximation algorithm, an exhaustive search strategy is adopted in MIC, that is, when the quadratic optimization is completed, other non-optimal partition sequences are also optimized [6]. Although this strategy improves the accuracy, it will increase time cost. In the trivariate associations analysis, as a comparison with QOTIC, we introduce the exhaustive search strategy and called it exhaustive search trivariate information coefficient (ESTIC).

**Definition** **6.**
*Trivariate equipartition information coefficient*

*For a data set D of three variables with sample size n, the trivariate equipartition information coefficient (TEIC) of D is given by*

(8)
$$\mathrm{TEIC}\left(D\right){=}_{xyz\leq B\left(n\right)}^{\;\;max}\left\{M^{E}{\left(D\right)}_{x,y,z}\right\}$$



The trivariate characteristic matrix and the trivariate equicharacteristic matrix have the same size of partition. TEIC is a variant of QOTIC which lacks the step to maximize over cubic grid partitions. TEIC simply uses the trivariate mutual information achieved by an adaptive equipartition at each cubic grid resolution, rather than considering all cubic grids at a given resolution and computing the maximal possible trivariate mutual information achieved by any of them. Relatively QOTIC contains a maximization step, which maximize a normalized variant of trivariate mutual information over a set of potential grids.

### 3.4. Time Complexity

The time cost of QOTIC includes two parts. The first part is the time cost of dynamically partition the *z*-axis, and the second part is the time cost of quadratic optimization of the *y*-axis. In the first part, given the upper bound *B* and parameter *c*, the partition numbers of *x*-axis and *y*-axis are *x* and *y,* respectively. The maximum number of dynamic partition of *z*-axis is *B*/(*xy*). Through dynamic partition of *z*-axis, the obtained cubic grid is partitioned into (*x*, *y*, 2), …, (*x*, *y*, *B*/(*xy*)), and the time complexity is $O\left({\left(cB/xy\right)}^{2}xy(B/xy)\right)=O\left(c^{2}B^{3}/{\left(xy\right)}^{2}\right)$. In the second part, fix the optimal partition sequence of *z*-axis with partition number of *B*/(*xy*) and *x*-axis partition number of *x*. The time complexity of quadratic optimization of *y*-axis is $O\left({\left(cy\right)}^{2}xyB/(xy)\right)=O\left(c^{2}y^{2}B\right)$.

For ESTIC, through dynamic partition of *z*-axis, the obtained cubic grid is partitioned into (*x*, *y*, 2), …, (*x*, *y*, *B*/(*xy*)), and the time complexity is $O\left({\left(cB/xy\right)}^{2}xy(B/xy)\right)=O\left(c^{2}B^{3}/{\left(xy\right)}^{2}\right)$. When the number of *z*-axis partition is 2, …, *B*/(*xy*), *y*-axis is optimized, respectively, and the time complexity is $O\left({\left(cy\right)}^{2}xy{\left(B/xy\right)}^{2}\right)=O\left(c^{2}B^{2}y/x\right)$.

The time complexity of ESTIC and QOTIC optimizing *y*-axis is $O\left(c^{2}B^{2}y/x\right)$ and, respectively. Because xy<B, $O\left(c^{2}y^{2}B\right)<O\left(c^{2}B^{2}y/x\right)$, and the time cost of dynamically partition *z*-axis of ESTIC and QOTIC is the same. So the time complexity of QOTIC is much less than that of ESTIC.

### 3.5. Mathematical Analysis

Let *m*^∞^ denote the space of infinite matrices equipped with the supremum norm. Given a trivariate matrix *A*∈*m*^∞^, for some *i*, only the *k*,*l,r*-th entries of *A* for which klr≤i are focused. Thus, for *i*∈*Z^+^*, we define the projection $r_{i}:m^{\infty }\to m^{\infty }$ via
(9)$$r_{i}{\left(A\right)}_{k,l,r}=\left\{ \begin{array}{l} A_{k,l,r}\;\;\;\;klr\leq i\\ 0\;\;\;\;\;\;\;\;\;\;klr>i \end{array}\right.$$

**Theorem** **1.***Let *$f:m^{\infty }\to R$* be uniformly continuous, and assume*$f\circ r_{i}\to f$* to be pointwise. Then for every three random variables (X, Y, Z), and a sample data set*D* of size n from the distribution of (X, Y, Z), provided *$\omega (1)<B(n)\leq O(n^{1-\varepsilon })$* for some*ε>0*, we have*(10)$$(f{}^{\circ}r_{B(n)})\left(M(D)\right)\to f\left(M\left(X,Y,Z\right)\right)\;$$*in probability*.

The theorem will be proved by the following sequence of lemmas that build on each other to bound the bias of I*(D,k,l,r). The general strategy is to capture the dependencies between different *k*-by-*l*-by-*r* cubic grids *G* by considering a master cubic grid Γ that contains much more than *klr* cubic cells. For the master cubic grid Γ, firstly the trivariate mutual information difference is bounded between $I((X,Y,Z){|}_{G})$ and $I(D{|}_{G})$ only for sub-grids *G* of Γ. Secondly, the boundary is extended to all *k*-by-*l-*by-*r* cubic grids without too much loss.

**Lemma** **1.**
*Let*

$\psi =(\psi_{X},\psi_{Y},\psi_{Z})$

* and*

$\varphi =(\varphi_{X},\varphi_{Y},\varphi_{Z})$

* be random variables distributed over the cells of a cubic grid *

Γ

*, and let (*

$\pi_{i,j,k}$

* ), (*

$\kappa_{i,j,k}$

*) be their respective distributions. Define*

(11)
$$\varepsilon_{i,j,k}=\frac{\kappa_{i,j,k}-\pi_{i,j,k}}{\pi_{i,j,k}}\;$$


*Let*
* G be a sub-grid of *

Γ

* with B cubic cells. Then for every fixed a, *0* < a < *1*, and index i, j and k, when *

$|\varepsilon_{i,j,k}|\leq 1-a$

*, we have*

(12)
$$\left|I(\varphi {|}_{G})-I(\psi {|}_{G})\right|\leq O\left(\log(B)\sum_{i,j,k}\left|\varepsilon_{i,jk}\right|\right)$$



**Proof.** Let *P* = $\psi {|}_{G}$, *Q* = $\varphi {|}_{G}$
be random variables induced by ψ
and φ, respectively, on the cells of cubic grid *G*. According to the theory of trivariate mutual information in Equation (1), we can get the following inequality.
(13)$$\begin{array}{ll} \left|I\left(Q\right)-I\left(P\right)\right|\leq & \left|H\left(Q_{X}\right)-H\left(P_{X}\right)\right|+\left|H\left(Q_{Y}\right)-H\left(P_{Y}\right)\right|+\left|H\left(Q_{Z}\right)-H\left(P_{Z}\right)\right|\\ & +\left|H\left(Q_{XY}\right)-H\left(P_{XY}\right)\right|+\left|H\left(Q_{XZ}\right)-H\left(P_{XZ}\right)\right|+\left|H\left(Q_{YZ}\right)-H\left(P_{YZ}\right)\right|\\ & +\left|H\left(Q\right)-H\left(P\right)\right| \end{array}$$
where *Q_X_* and *P_X_* denote the marginal distributions along the *x*-axis of *G*. *Q_Y_* and *P_Y_* denote the marginal distributions along the *y*-axis of *G*. *Q_Z_* and *P_Z_* denote the marginal distributions along the *z*-axis of *G.* Similarly, $Q_{XY},\;P_{XY},\;Q_{XZ},\;P_{XZ},\;Q_{YZ},\;P_{YZ}$ denote the marginal distributions along the *x*-axis, *y*-axis of *G, x*-axis, *z*-axis of *G* and *y*-axis, *z*-axis of *G*, respectively. We bound each of the terms on the right side of the equation above using a Taylor Expansion Argument [11]. Accordingly, we get
(14)$$\left|I\left(Q\right)-I\left(P\right)\right|\leq \left(\log B\right)\left(\begin{array}{l} \sum_{i}\varepsilon_{i,*,*}+\sum_{j}\varepsilon_{*,j,*}\sum_{k}\varepsilon_{*,*,k}+\sum_{i,j,k}\varepsilon_{i,j,k}\\ +\sum_{i,j}\varepsilon_{i,j,*}+\sum_{i,k}\varepsilon_{i,*,k}+\sum_{j,k}\varepsilon_{*,j,k} \end{array}\right)$$
where
$$\varepsilon_{i,*,*}=\frac{\sum_{j,k}\left(\kappa_{i,j,k}-\pi_{i,j,k}\right)}{\sum_{j,k}\pi_{i,j,k}},\;\varepsilon_{*,j,*}=\frac{\sum_{i,k}\left(\kappa_{i,j,k}-\pi_{i,j,k}\right)}{\sum_{i,k}\pi_{i,j,k}},\;\varepsilon_{*,*,k}=\frac{\sum_{j,k}\left(\kappa_{i,j,k}-\pi_{i,j,k}\right)}{\sum_{j,k}\pi_{i,j,k}}$$
$$\varepsilon_{i,j,*}=\frac{\sum_{k}\left(\kappa_{i,j,k}-\pi_{i,j,k}\right)}{\sum_{k}\pi_{i,j,k}},\;\varepsilon_{i,*,k}=\frac{\sum_{j}\left(\kappa_{i,j,k}-\pi_{i,j,k}\right)}{\sum_{j}\pi_{i,j,k}},\;\varepsilon_{*,j,k}=\frac{\sum_{i}\left(\kappa_{i,j,k}-\pi_{i,j,k}\right)}{\sum_{i}\pi_{i,j,k}}$$From the above, we can obtain the following derivation,
$$\left|\varepsilon_{i,*,*}\right|=\left|\frac{\sum_{j,k}\pi_{i,j,k}\varepsilon_{i,j,k}}{\sum_{j,k}\pi_{i,j,k}}\right|\leq \frac{\sum_{j,k}\pi_{i,j,k}\left|\varepsilon_{i,j,k}\right|}{\sum_{j,k}\pi_{i,j,k}}\leq \sum_{j,k}\left|\varepsilon_{i,j,k}\right|$$
$$\left|\varepsilon_{i,j,*}\right|=\frac{\sum_{k}\pi_{i,j,k}\varepsilon_{i,j,k}}{\sum_{k}\pi_{i,j,k}}\leq \frac{\sum_{k}\pi_{i,j,k}\left|\varepsilon_{i,j,k}\right|}{\sum_{k}\pi_{i,j,k}}\leq \sum_{k}\left|\varepsilon_{i,j,k}\right|$$
since $\pi_{i,j,k}/\sum_{j,k}\pi_{i,j,k}\leq 1$, $\pi_{i,j,k}/\sum_{k}\pi_{i,j,k}\leq 1$. Analogous bound holds for $\left|\varepsilon_{*,j,*}\right|$, $\left|\varepsilon_{*,*,k}\right|$, $\left|\varepsilon_{*,j,k}\right|$ and $\left|\varepsilon_{i,*,k}\right|$. Therefore, |I(Q)−I(P)|≤$O\left(\log(B)\sum_{i,j,k}\left|\varepsilon_{i,j,k}\right|\right)$ is proved. □

**Lemma** **2.***Define random variables*$\psi =(\psi_{X},\psi_{Y},\psi_{Z})$*,*$\varphi =(\varphi_{X},\varphi_{Y},\varphi_{Z})$* as in Lemma 1, and *$\psi {|}_{\Gamma }$*,*$\varphi {|}_{\Gamma }$ *be random variables induced by*ψ *and*φ*,**respectively, on the cells of master cubic grid*Γ.* Let G be any cubic grid with B cells, and let *δ *represent the total probability mass of*$\psi {|}_{\Gamma }$ *falling in cells of*Γ *which are not contained in individual cells of G, and let d represent the total probability mass of*$\varphi {|}_{\Gamma }$ *falling in cells of *Γ *which are not contained in individual cells of G. If*δ,d≤1/2, *and *$\left|\varepsilon_{i,j,k}\right|$*are bounded away from 1, the following inequality holds*


(15)
$$\left|I(\varphi {|}_{G})-I(\psi {|}_{G}\right)|\leq O\left(\left(\sum_{i,j,k}\left|\varepsilon_{i,j,k}\right|+\delta +d)\log B+\delta \log(1/\delta )+d\log(1/d\right)\right)\;$$


**Proof.** For any two cubic grids *G* and *G*′, grid *G*′ is obtained by replacing every dividing plane in *G* which is not in Γ with a closest line in Γ. Obviously, *G*′ is a sub-grid of Γ. Let $\zeta \;=\;(\zeta_{X},\zeta_{Y},\zeta_{Z})$ be random variables, and $\zeta {|}_{G}$, $\zeta {|}_{G^{\prime }}$ are random variables induced by ζ on the cells of cubic grid *G* and *G*′, respectively. The absolute value of trivariate mutual information difference of $\zeta {|}_{G}$, $\zeta {|}_{G^{\prime }}$ is expressed as $\Delta^{\zeta }(G,G^{\prime })=\left|I(\zeta {|}_{G})-I(\zeta {|}_{G^{\prime }})\right|$. For ψ, $\psi {|}_{G^{\prime }}$ can be obtained from $\psi {|}_{G}$ by moving at most δ probability mass. For φ, $\varphi {|}_{G^{\prime }}$ can be obtained from $\varphi {|}_{G}$ by moving at most *d* probability mass. From the information-theoretic fact [11].
(16)$$\begin{array}{c} \Delta^{\psi }(G,G^{\prime })\leq O\left(\delta \log(1/\delta )+\delta \log(B)\right),\;\\ \Delta^{\varphi }(G,G^{\prime })\leq O{\left(d\log(1/d)+d\log(B)\right)}^{\;} \end{array}$$Since $\Delta^{\psi }(G,G^{\prime })=|I(\psi {|}_{G})-I(\psi {|}_{G^{\prime }})|$, $\Delta^{\varphi }(G,G^{\prime })=|I(\varphi {|}_{G})-I(\varphi {|}_{G^{\prime }})|$, then with $\Delta^{\psi }(G,G^{\prime })$ and $\Delta^{\varphi }(G,G^{\prime })$ bounded in terms of δ and *d*, $|I(\varphi {|}_{G})-I(\psi {|}_{G})|$ can also be bounded by using the triangle inequality,
$$\begin{array}{ll} \Delta^{\psi }(G,G^{\prime })+\Delta^{\varphi }(G,G^{\prime }) & \geq |(I(\psi {|}_{G})-I(\psi {|}_{G^{\prime }}))-(I(\varphi {|}_{G})-I(\varphi {|}_{G^{\prime }}))|\\ & =|(I(\psi {|}_{G})-I(\varphi {|}_{G}))+(I(\varphi {|}_{G^{\prime }})-I(\psi {|}_{G^{\prime }}))|\\ & \geq |(I(\psi {|}_{G})-I(\varphi {|}_{G}))|-|(I(\varphi {|}_{G^{\prime }})-I(\psi {|}_{G^{\prime }}))|\\ \Rightarrow \;|(I(\psi {|}_{G})-I(\varphi {|}_{G}))| & \leq \Delta^{\psi }(G,G^{\prime })+\Delta^{\varphi }(G,G^{\prime })+|(I(\varphi {|}_{G^{\prime }})-I(\psi {|}_{G^{\prime }}))|\\ \Rightarrow \;|I(\varphi {|}_{G})-I(\psi {|}_{G})|\leq & O(\left(\sum_{i,j,k}\left|\varepsilon_{i,j,k}\right|+\delta +d)\log B+\delta \log(1/\delta )+d\log(1/d\right)) \end{array}□$$

**Lemma** **3.** *Let D be a sample of size n from the distribution of a pair (X, Y, Z) of jointly distributed random variables. For *α≥0,ε>0*, and any k-by-l-by-r cubic grid G, we have*(17)$$\left|I(D{|}_{G})-I((X,Y,Z){|}_{G}\right)|\leq O\left(\frac{\log(klr)}{C{\left(n\right)}^{\alpha }}+\frac{\log(klrn)}{n^{\varepsilon /9}}\right)\;$$*with probability at least*$1-C(n)e^{-O\left(n/C{\left(n\right)}^{1+3\alpha }\right)}$*, where*$C(n)=klrn^{\varepsilon /3}$.

**Proof.** Fix a sample size *n*, and let Γ be a cubic grid which makes an equipartition of (*X*, *Y*, *Z*) into $kn^{\varepsilon /8}$ bins along the *x*-axis, $ln^{\varepsilon /8}\;$ bins along the *y*-axis and $rn^{\varepsilon /8}\;$
 bins along the *z*-axis. Then *C*(*n*) represents the total number of cubic cells. From Lemma 2, with ψ=(X, Y, Z) and φ=D, shows that $\left|I(D{|}_{G})-I((X,Y,Z){|}_{G}\right)|$ is at most
(18)$$|I(\varphi {|}_{G})-I(\psi {|}_{G})|\leq O\left(\left(\sum_{i,j,k}\left|\varepsilon_{i,j,k}\right|+\delta +d)\log B+\delta \log(1/\delta )+d\log(1/d\right)\right)$$We bound the $\varepsilon_{i,j,k}$ using a multiplicative Chernoff bound first, let $\pi_{i,j,k}$ and $\kappa_{i,j,k}$ represent the probability mass functions of $(X,Y,Z){|}_{\Gamma }$ and $D{|}_{\Gamma }$, respectively. From solving absolute value inequality, we can obtain that $P(\left|\varepsilon_{i,j,k}\right|\geq \delta )=P(\kappa_{i,j,k}\geq \pi_{i,j,k}(1+\delta ))$ or $P(\left|\varepsilon_{i,j,k}\right|\geq \delta )=P(\kappa_{i,j,k}\leq \pi_{i,j,k}(1-\delta ))$. Since $\kappa_{i,j,k}$ is a sum of *n* independent and identically distributed (i.i.d) Bernoulli random variables, and $E(\kappa_{i,j,k})=n\pi_{i,j,k}$, then there is $P(\left|\varepsilon_{i,j,k}\right|\geq \delta )\leq e^{-\Omega (n\pi_{i,j,k}\delta^{3})}$. Setting δ = ${\pi_{i,j,k}}^{1/3}/C{\left(n\right)}^{1/3+\alpha }$, yields
(19)$$P(\left|\varepsilon_{i,j,k}\right|\geq \frac{{\pi_{i,j,k}}^{1/3}}{C{\left(n\right)}^{1/3+\alpha }})\leq e^{-O\left(n/C{\left(n\right)}^{1+3\alpha }\right)}$$According to the probability results above, when $\pi_{i,j,k}<1$, C(n)>1, a union bound over all index pairs (*i*, *j*, *k*) is
(20)$$\begin{array}{ll} \sum_{i,j,k}\left|\varepsilon_{i,j,k}\right| & \leq \frac{1}{C{\left(n\right)}^{1/3+\alpha }}\sum_{i,j,k}{\pi_{i,j,k}}^{1/3}\\ & \leq \frac{1}{C{\left(n\right)}^{1/3+\alpha }}C{\left(n\right)}^{\frac{1}{3}}\\ & \leq \frac{1}{C{\left(n\right)}^{\alpha }} \end{array}$$From this, we can get the relationship between $\kappa_{i,j,k}$ and $\pi_{i,j,k}$,
(21)$$\begin{array}{ll} \kappa_{i,j,k} & \leq \pi_{i,j,k}(1+\delta )=\pi_{i,j,k}(1+\frac{{\pi_{i,j,k}}^{1/3}}{C{\left(n\right)}^{1/3+\alpha }})\\ & =\pi_{i,j,k}+\frac{{\pi_{i,j,k}}^{4/3}}{C{\left(n\right)}^{1/3+\alpha }}\\ & \leq \pi_{i,j,k}+\frac{\pi_{i,j,k}}{C{\left(n\right)}^{1/3+\alpha }}\\ & \leq 2\pi_{i,j,k} \end{array}$$The connection to *d* comes from the fact that for any bin *k* along *z*-axis of Γ, which means that $\kappa_{*,*,k}=\sum_{i,j}\kappa_{i,j,k}\leq 2\sum_{i,j}\pi_{i,j,k}=2\pi_{*,*,k}$. Similarly, this also applies to the sums across bins along *x*-axis and bins along *y*-axis, thus, $\kappa_{i,*,*}\leq 2\pi_{i,*,*}$, $\kappa_{*,j,*}\leq 2\pi_{*,j,*}$.Since *d* is a sum of terms of the form $\kappa_{i,*,*}$, $\kappa_{*,j,*}$ and $\kappa_{*,*,k}$ for *i* in some index set *I*, *j* in an index set *J* and *k* in some index set *K*. Similarly, δ is a sum of terms of the form $\pi_{i,*,*}$, $\pi_{*,j,*}$ and $\pi_{*,*,k}$ with the same index sets. Therefore, *d* is bounded in terms of δ, d≤2δ. Because *G* has at most *k* bins along *x*-axis, *l* bins along *y*-axis and *r* bins along *z*-axis, then the inequality can be obtained as follows.
$$\delta \leq \frac{l}{ln^{\varepsilon /9}}+\frac{k}{kn^{\varepsilon /9}}+\frac{r}{rn^{\varepsilon /9}}\leq \frac{3}{n^{\varepsilon /9}}$$
$$\delta +d\leq O(\frac{1}{n^{\varepsilon /9}}),\;\delta \log(\frac{1}{\delta })+d\log(\frac{1}{d})\leq O(\frac{\log n}{n^{\varepsilon /9}})$$Combining all of the bounds gives the desired result. □

**Lemma** **4.** *Let D be a sample of size n from the distribution of a pair (X, Y, Z) of jointly distributed random variables, for every*$B(n)=O(n^{1-\varepsilon })$, *such that for large enough n,*(22)$$\left|M{\left(D\right)}_{k,l,r}-M{\left(X,Y,Z\right)}_{k,l,r}\right|\leq O\left(\frac{1}{n^{a}}\right)\;$$*holds for all*klr≤B(n)*with probability*P(n)=1−o(1).
*Where*

$\;M{\left(D_{n}\right)}_{k,l,r}$

* is the *
*k, l, r-th entry of the sample trivariate characteristic matrix,*

$\;M{\left(X,Y,Z\right)}_{k,l,r}$

* is the *
*k, l, r-th entry of the population trivariate characteristic matrix.*


**Proof.** For fixed *k*, *l* and *r*, according to Lemma 3, it implies that with high probability the difference $M{\left(D\right)}_{k,l,r}-M{\left(X,Y,Z\right)}_{k,l,r}$ is at most
$$O\left(\frac{\log(klr)}{C{\left(n\right)}^{\alpha }}+\frac{\log(klrn)}{n^{\varepsilon /9}}\right)\leq O\left(\frac{\log n}{C{\left(n\right)}^{\alpha }}+\frac{\log n}{n^{\varepsilon /9}}\right)\leq O\left(\frac{\log n}{n^{\alpha \varepsilon /3}}+\frac{\log n}{n^{\varepsilon /9}}\right)$$Since $C(n)=klrn^{\varepsilon /3}\geq n^{\varepsilon /3}$,  klr≤B(n), for every a≤min{αε/3,ε/9}, this boundary is at most $O(1/n^{a})$. It is only to show that the boundary holds with high probability across all *klr* < *B*(*n*). For some *u >* 0, and a large sample size *n*, $C(n)\leq B(n)n^{\varepsilon /3}\leq O(n^{1-\varepsilon /3})$, because the choice of α ensures that $C{\left(n\right)}^{1+3\alpha }=O(n^{1-u})$ for some *u*, the probability of our bound holds is at least $1-C(n)e^{-O\left(n/C{\left(n\right)}^{1+3\alpha }\right)}\geq 1-O(n)e^{-O\left(n^{u}\right)}$. Then perform the boundary over all  klr≤B(n), the desired condition is satisfied with probability approaching 1. □

**Proof of Theorem 1:** Let $M_{B(n)}=r_{B(n)}(M)$, and let $M_{B(n)}\left(D\right)=r_{B(n)}(M\left(D\right))$, then we have the following inequality
$$\begin{array}{ll} \left|f\left(M_{B(n)}(D)\right)-f(M)\right| & \leq \left|f\left(M_{B(n)}(D)\right)-f(M_{B(n)})\right|+\left|f(M_{B(n)})-f(M)\right|\\ & =\left|f\left(M_{B(n)}(D)\right)-f(M_{B(n)})\right|+\left|f{}^{\circ}r_{B(n)}(M)-f(M\right)| \end{array}$$The second term on right-hand side of the inequality vanishes by the pointwise convergence of $f{}^{\circ}r_{i}$ as n→∞. From the fact that B(n)>ω(1), therefore it is enough to show that the first term converges to 0 in probability.Let ‖ ⋅ ‖ denote the supremum norm on $m^{\infty }$, and fix any *z >* 0, then for any δ>0, define
$$C_{\delta }=\left\{A\in m^{\infty }:\exists A^{\prime }\in m^{\infty }\;s.t\left\|A-A^{\prime }\right\|<\delta ,\left|f(A)-f(A^{\prime })\right|>z\right\}$$This is a set of matrices $A\in m^{\infty }$ which is possible to be found, within a δ- neighborhood of *A*, a second matrix $A^{\prime }$ that *f* maps to more than *z* away from *f*(*A*). Because *f* is uniformly continuous, there exists a small enough $\delta^{*}$, to make $C_{\delta^{*}}=\emptyset$.Supposing $\left|f\left(M_{B(n)}(D)\right)-f(M_{B(n)})\right|>z$, which means that either $\left\|M_{B(n)}(D)-M_{B(n)}\right\|>\delta^{*}$ or $M_{B(n)}\in C_{\delta^{*}}$. The latter item is impossible since $C_{\delta^{*}}=\emptyset$, and from Lemma 4, with the increase of sample size *n*, $P\left(\left\|M_{B(n)}(D)-M_{B(n)}\right\|>\delta^{*}\right)\to 0$. Thus, we have that $\left|f\left(M_{B(n)}(D)\right)-f(M_{B(n)})\right|\to 0$ in probability, as expected. □

## 4. Comparisons with Other Methods

We experimentally compare different noise levels in various functions to contrast the performance of QOTIC with the baseline methods. In the experiment, first noiseless relationships are used to test the generality and time cost, and then noisy relationships are employed to test the equitability. 

### 4.1. Performance on Noiseless Relationships

In order to verify the performance of QOTIC in generality, 12 functional relationships as shown in Figure 6 are used, in which the sample sizes are 500 and 1000, respectively. All relationships are uniformly distributed and noiseless. Take MTDIC and ESTIC, which are based on the approximation algorithm, as comparison. The parameter settings of the three methods MTDIC, QOTIC and ESTIC are the same (*α* = 0.75, *c* = 10). The experimental hardware conditions are CPU Intel i5 and memory 8.00 GB. Table 3 shows the results of the correlation values given for the 12 functional relationships.

In Figure 6, given an arbitrary random variable *r*, three variables on each axis are *X* = *r*, *Y* = *f* (*r*), and *Z* = *f* (*r*). *f* corresponds to one of the 12 functional relationships.

As shown in Table 3, firstly, QOTIC and ESTIC have the same accuracy, and the correlation values given for all relationships are the same. Secondly, for the three monotonic relationships linear, exponential, logarithmic and step function, when the sample size is 500 and 1000, respectively, MTDIC, QOTIC, and ESTIC all give the highest correlation value 1. For the remaining eight relationships, overall, with the increase of sample size, the correlation values given by MTDIC, QOTIC and ESTIC are increasing. When the sample size of quadratic is 500, the correlation values given by QOTIC and ESTIC are higher than those given by MTDIC. When the sample size increases to 1000, MTDIC, QOTIC and ESTIC all give the highest correlation value of 1. The correlation values given by MTDIC, QOTIC and ESTIC for two lines are equal. For the other six relationships, the correlation values given by QOTIC and ESTIC are higher than those given by MTDIC.

Time cost analysis of three methods MTDIC, ESTIC and QOTIC is based on the 12 functional relationships shown in Figure 6. Firstly, analyze the time cost on each relationship with sample size 500, in seconds (s). The results are shown in Table 4. Secondly, analyze the time cost on all relationships with sample size 100 to 1000. The results are shown in Figure 7.

According to Table 4, in general, for each functional relationship, the lowest time cost of the three methods is MTDIC and the highest time cost is ESTIC. MTDIC has large time cost differences in 12 relationships. For three monotonic relationships, linear, exponential and logarithmic, the time cost difference is small, and all of them are within 0.1 s. The time cost on quadratic, sinusodial low frequency, sinusodial high frequency, circle, X line and X curve is higher, all of which are more than 3 s. There is no difference in the time cost between ESTIC and QOTIC in 12 functional relationships, which are 5.13 s and 7.13 s, respectively. QOTIC is consistent with ESTIC with the highest accuracy, and the time loss is reduced by 28%.

As shown in Figure 7, when the sample size is less than 300, the time cost difference of the three methods is small, and time cost are all within 15 s. When the sample size is larger than 300, the time cost gap of the three methods begins to expand with the increase of the sample size. First, the MTDIC with the lowest time cost increases slowly with the increase of sample size. When the sample size reaches 1000, the time cost is still less than 13 s. ESTIC with the highest time cost, when the sample size is larger than 300, the time cost increases exponentially with the increase of the sample size. When the sample size of QOTIC is higher than 300, although the time cost increases rapidly, it is much lower than ESTIC, and the time cost gap between QOTIC and ESTIC increases with the increase of sample size.

### 4.2. Performance on Noisy Relationships

The noisy relationships refer to 12 different functions shown in Table 1. Each relationship uses the sample size of 100, 500, and 1000, respectively. Each functional relationship is also uniformly distributed and contains additive Gaussian noise. The noise level increases gradually, and the coefficient of determination *R*^2^ ranges from 0 to 1.

Figure 8 shows the equitability results of QOTIC and the other five baseline methods, Dcor, MIE, NCIE, TEIC, and MTDIC. Each subplot contains the score of the given statistic versus the coefficient of determination *R*^2^. 

In Figure 8, QOTIC assigns similar scores to these functions under the same noise levels, and when the sample size increases, QOTIC roughly equals the coefficient of determination *R*^2^. Methods, such as MTDIC and MIE, tend to be equitable across these functions, but MIE is not sensitive to the sample size. The performance of equitability is not improved with the increase of sample size. Moreover, under different sample sizes, the equitability performance of MTDIC and MIE are not as well as QOTIC. For the other methods TEIC, Dcor and NCIE all detect broader classes of relationships, but they are not equitable even in the large sample size of functions.

## 5. Exploring GHO Dataset for Associations

The Global Health Observatory (GHO) dataset is explored by our proposed QOTIC and the existing MIC, which is collected from the World Health Organization (WHO) data repository website and the United Nations website. The dataset consists of 2938 samples in 18 columns documenting the life expectancy and the corresponding potential influence factors such as immunization factors, mortality factors, social factors, economic factors and other health-related factors, ranging from year 2000 to 2015 around 193 countries. In trivariate correlation detection, each variable is arbitrarily combined with the other two variables, and then use QOTIC to measure correlation. As for the original correlation to explore, each variable is arbitrary combined with another variable, and then use MIC to measure correlation, each set of trivariate correlation includes three groups of relations between the two bivariate correlation, the relationship between trivariate association and bivariate association is shown in Table 5.

### 5.1. Trivariate Associations with QOTIC

The 18 factors are first combined in pairs, and then the relationships between $C_{18}^{2}=153$ pairwise factors and life expectancy are explored by QOTIC (see Figure 9).

Figure 9a shows the histogram of QOTIC scores. Among them, there are only 20 pairwise factors that are strongly related to life expectancy. The medium-strong relationships with the scores between 0.4 and 0.6 account for 0.59%. The weak relationships with the scores between 0.2 and 0.4 account for the largest proportion, 48.4%, and there are 50 extreme-weak relationships with the scores between 0.2 and 0.4. Therefore, it is not rare for pairwise factors to have an association with life expectancy, but there are relatively few strong associations. Furthermore, we select several representative relationships with strong associations. Many of these have not been in the existing literatures because there are few measures to detect the trivariate variable pairs in GHO data set.

Figure 9b,c show that two pairwise mortality factors (infant deaths and under-five deaths), thinness 1–19 years, and thinness five to nine years have a strong correlation with life expectancy. The correlation values of the two relationships are 0.942 and 0.857, respectively. Where Figure 9b shows an obvious negative correlation between young child mortality and life expectation, and from Figure 9c, the relationship between thinness of younger group and life expectancy is also a negative correlation.

In Figure 9d, the health-related factors of polio and diphtheria represent the immunization coverage among 1-year-olds. It is easy to see that there is a significant positive correlation between the immunization factor of two diseases and life expectancy, and the correlation value is 0.777.

Figure 9e,f indicate two relationships on economic and social factors, and their corresponding correlation values are 0.685 and 0.67, respectively. For Figure 9e, life expectancy increases as expenditure on health and GDP increase, and Figure 9f, shows that life expectancy increases linearly with the increase of factor indexes of income and education.

The last two relationships in Figure 9g,h are very similar, involving the mortality and economic factors, and their correlation values are 0.642 and 0.64, respectively. For Figure 9g, the relationship identified is not a simple functional association between life expectancy and factor indexes of adult mortality and expenditure on health, and it illustrates a superposition of multiple functions. Similarly, the superposition of multiple functional associations between the life expectation and the factor indexes of GDP and adult mortality is shown in Figure 9h.

### 5.2. Bivariate Associations with MIC

19 variables include 18 factors and life expectancy generating $C_{19}^{2}=171$ pairwise variables in GHO dataset. These 171 pairwise variables are explored by MIC (see Figure 10).

Figure 10a shows the histogram of MIC scores. Among them, there are only 7 strong relationships with the scores between 0.6 and 0.1. The medium-strong relationships with the scores between 0.4 and 0.6 account for 7.6%. The weak relationships with the scores between 0.2 and 0.4 account for the largest proportion, 57.9%, and there are 52 extreme-weak relationships with the scores between 0.2 and 0.4. From the histogram of MIC scores, among these 19 variables, it is not rare for one variable to have an association with any other variable, but strong associations are relatively few. For the seven strong relationships, they are ranked by MIC score and shown in Figure 10b–h.

Figure 10b,c indicate the positive correlation, and the two relationships are obviously linear with a slight noise. The correlation value of the two relationships given by MIC are 0.958 and 0.912, respectively. Figure 10b demonstrates the relationship between infant deaths and under-five deaths, and the points of relationship presents an exponential distribution. Figure 10c demonstrates the relationship between thinness five to nine years and thinness 1–19 years.

Figure 10d presents the association of the health-related factors between polio and diphtheria. It is obvious that the relationship of the two factors is complex, but it generally illustrates a positive correlation and correlation value given by MIC is 0.801.

Figure 10e,f show two relationships which involve economic and social factors, and the corresponding correlation value are 0.72 and 0.714, respectively. Figure 10e demonstrates a rough positive correlation between income composition of resources and schooling. Figure 10f demonstrates the positive correlation between the percentage of expenditure and GDP, the points of relationship present an exponential distribution with a lot of noise.

The last two relationships in Figure 10g,h demonstrates two factors that have a strong association with life expectancy, and their correlation values are 0709 and 0.614, respectively. Figure 10g illustrates an obvious negative correlation between adult mortality and life expectancy, while Figure 10h illustrates the positive correlation between income composition of resources and life expectancy. Besides, the relationships shown in Figure 10e,g are very similar, both rough positive correlations.

### 5.3. Comparing QOTIC with MIC on GHO Dataset

For GHO dataset, the trivariate associations on trivariate variables are detected by QOTIC in Section 5.1, and the bivariate associations on pairwise variables are detected by MIC in Section 5.2. In this section, these two results will be made further comparative analysis on the performance of QOTIC and MIC. Table 5 shows the correlation value of the trivariate variable and the correlation value of the binary variable. 

In Table 5, there are 7 groups compared case, and each trivariate association corresponds to three bivariate associations.

The first group of trivariate association is infant deaths, under-five deaths and life expectancy, and its correlation value is 0.942. In the three sets of bivariate associations, the correlation values of infant deaths, under-five deaths and life expectancy are 0.31, 0.342, and the correlation value of infant deaths and under-five deaths is 0.958.The second group of trivariate association is thinness 1–19 years, thinness five to nine years and life expectancy, and its correlation value is 0.857. In the three sets of bivariate associations, the correlation values of thinness 1–19 years, thinness five to nine years and life expectancy are 0.386 and 0.384, respectively, while the correlation values of thinness 1–19 years and thinness five to nine years are 0.912.The third group of trivariate association is polio, diphtheria and life expectancy, and its correlation value is 0.777. In the three sets of bivariate associations, the correlation values of polio, diphtheria and life expectancy are 0.298, 0.295, while the correlation value of polio and diphtheria is 0.801.The fourth group of trivariate association is the percentage of expenditure, GDP and life expectancy, and its correlation value is 0.685. In the three sets of bivariate associations, the correlation values of the percentage of expenditure, GDP and life expectancy are 0.31 and 0.377, respectively, while the correlation value of the percentage of expenditure and GDP is 0.714.The fifth group of trivariate association is income composition of resources, schooling and life expectancy, and its correlation value is 0.67. In the three sets of bivariate associations, the correlation values of income composition of resources, schooling and life expectancy are 0.614 and 0.497, respectively, while the correlation value of percentage expenditure and GDP is 0.72.The sixth group of trivariate associations is Adult Mortality, percentage expenditure and life expectancy, and its correlation value is 0.642. In the three sets of bivariate associations, the correlation values of adult mortality, the percentage of expenditure and life expectancy are 0.709 and 0.31, respectively, while the correlation values of adult mortality and the percentage of expenditure are 0.22.The seventh group of trivariate associations is adult mortality, GDP and life expectancy, and its correlation value is 0.64. In the three sets of bivariate associations, the correlation values of adult mortality, GDP and life expectancy are 0.709 and 0.377, respectively, while the correlation value of adult mortality and GDP is 0.284.

The seven groups of trivariate associations are all of the strong correlations in Section 5.1. By comparing the correlations with binary variables, these seven groups of trivariate associations can be divided into three categories. The first category is the first to the fourth group. In this category, the relationship between each factor and life expectancy binary variables is weak, and the relationship between factor and factor is strong. The second category is the fifth group. One factor has a strong correlation with the binary variable of life expectancy, while the other factor has a weak correlation with the binary variable of life expectancy, and the binary variable between factor and factor has a strong correlation. The third category is the sixth and seventh groups. In this category, one factor has a strong correlation with the life expectancy binary variable, and the other factor has a weak correlation with the life expectancy binary variable. For the factor and the factor, the correlation between the binary variables is weak.

Moreover, for the two factors, although the bivariate associations formed by their respective combination with life expectancy is weakly correlated, the trivariate associations between the two factors and life expectancy is indeed strongly correlated. This also means that in the analysis of the impact of multiple factors on the target variable, if you only rely on the strength of the correlation between the binary variables to screen the factors, it may be discarded because of the weak correlation between the single factor and the target variable, ignoring the influence of other factors on the target variable combined with it.

## 6. Conclusions and Future Works

It is important to detect the strong associations from complex relationships in high-dimensional datasets. Although MIC measures the correlation of pairwise variables, few methods can satisfy the generality and equitability for measuring the correlation among three variables. In this paper, quadratic optimized trivariate information coefficient (QOTIC) was proposed to measure the correlation among three variables. Based on the principles of trivariate information, we presented a quadratic optimization on two axes to encapsulate the relationships existing in the three dimensions. For comparison, as a variant of QOTIC, trivariate equipartition information coefficient (TEIC) uses an adaptive equipartition on three axes and lacks the maximum step. The comparison of equitability show that the dynamic partition is noticeably better than that of the adaptive equipartition, and the accurate correlation can be reflected only when the relationship is suitably encapsulated. Furthermore, QOTIC performed the best in the equitability and generality when compared with other methods by using different functional relationships, different noises, and different sample sizes, especially the sample size is large enough. Finally, we applied QOTIC to a real-world data set to explore the trivariate associations. The results show that QOTIC can effectively detect various relationships. 

QOTIC also has limitations. QOTIC is only applicable to trivariate variables, but the association from higher dimensions cannot be mined. In addition, QOTIC selects the maximum value in the trivariate characteristic matrix as the correlation measure, so the trivariate variable relationship containing noise may be assigned the maximum correlation value, resulting in overestimation. Future studies will focus on the potential overestimation of QOTIC correlation and expect to propose a method to detect and correct the overestimation relationship. In addition, it is hoped that the use of QOTIC can be extended from trivariate to multivariate so as to explore the potential association of multivariate variables.

## Figures and Tables

**Figure 1 sensors-22-02806-f001:**
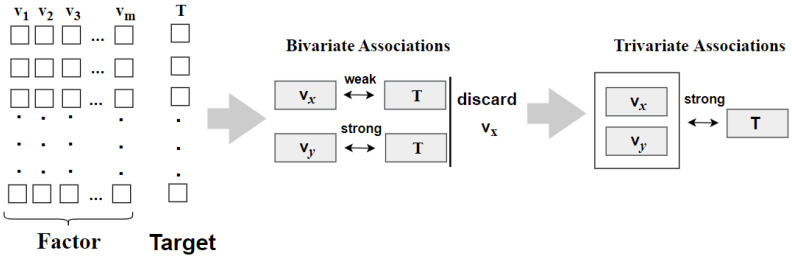
Advantages of trivariate associations analysis.

**Figure 2 sensors-22-02806-f002:**
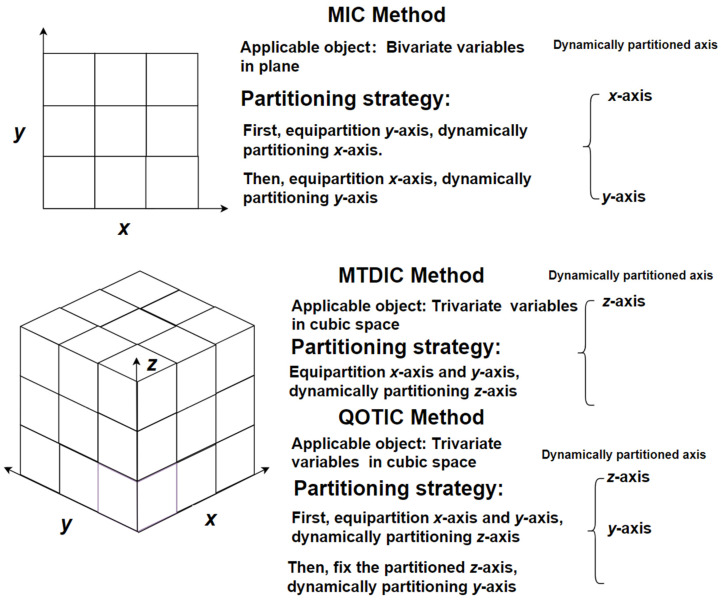
Partition strategies of bivariate variables and trivariate ones.

**Figure 3 sensors-22-02806-f003:**
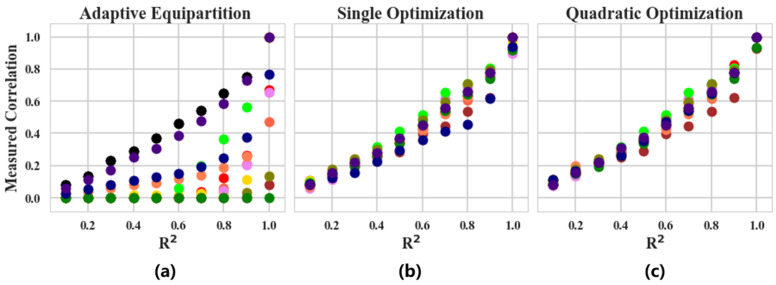
Comparison of equitability of adaptive equipartition, single optimization and quadratic optimization.

**Figure 4 sensors-22-02806-f004:**
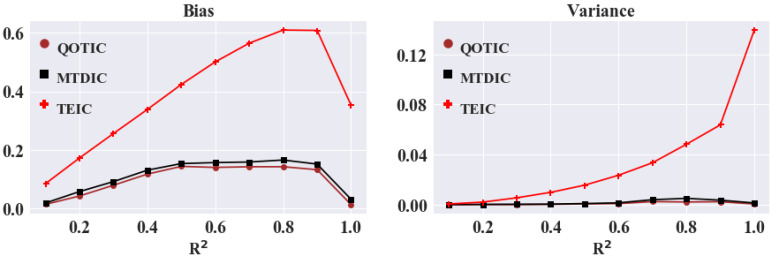
Comparison of bias and variance of QOTIC, MTDIC, TEIC.

**Figure 5 sensors-22-02806-f005:**
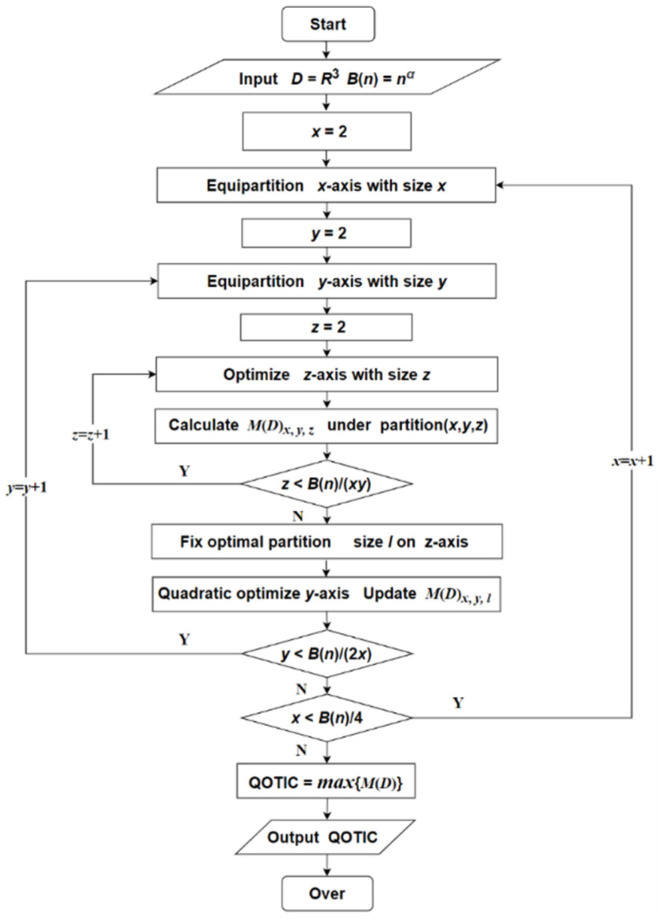
The flow chart of QOTIC method.

**Figure 6 sensors-22-02806-f006:**
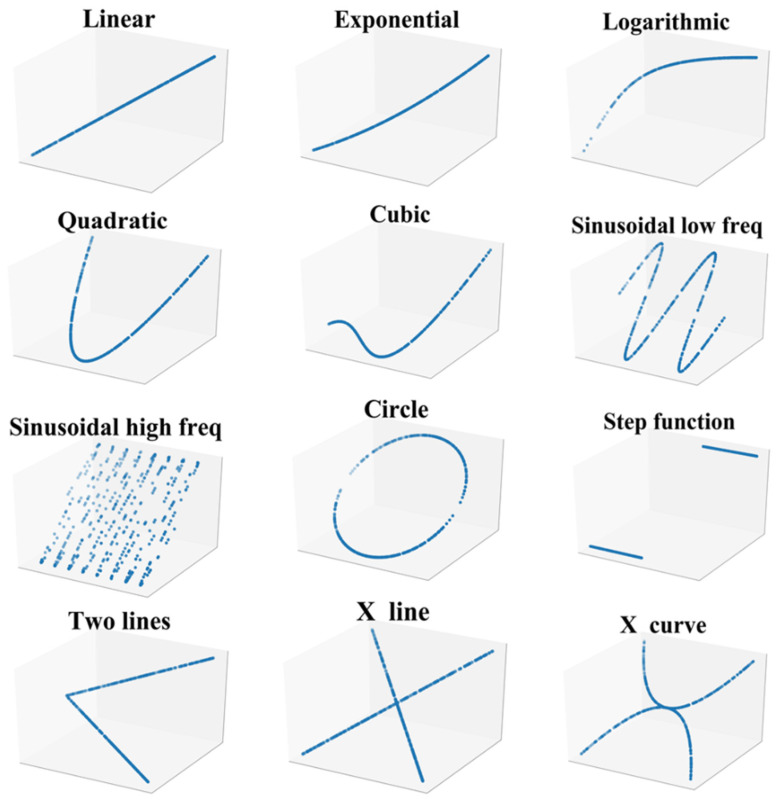
12 functional relationships used to verify generality.

**Figure 7 sensors-22-02806-f007:**
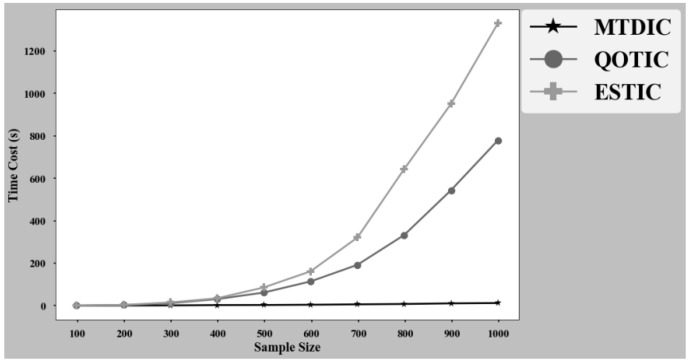
Time cost with different sample sizes.

**Figure 8 sensors-22-02806-f008:**
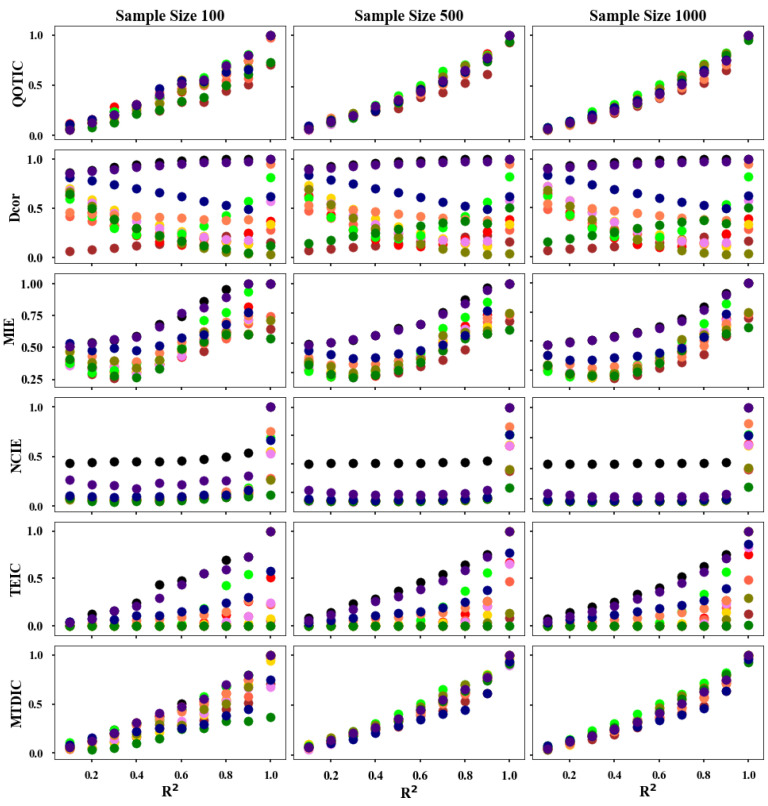
Equitability comparison of six trivariate correlation methods.

**Figure 9 sensors-22-02806-f009:**
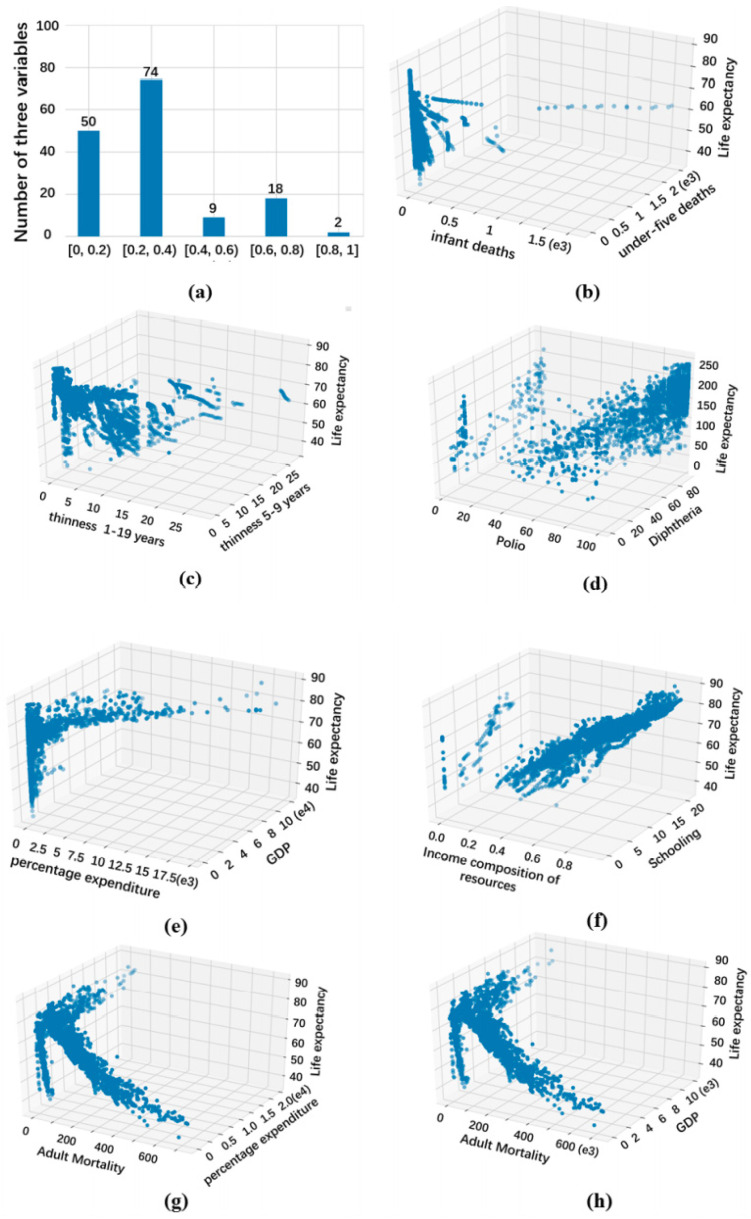
Exploring GHO dataset for trivariate associations with QOTIC.

**Figure 10 sensors-22-02806-f010:**
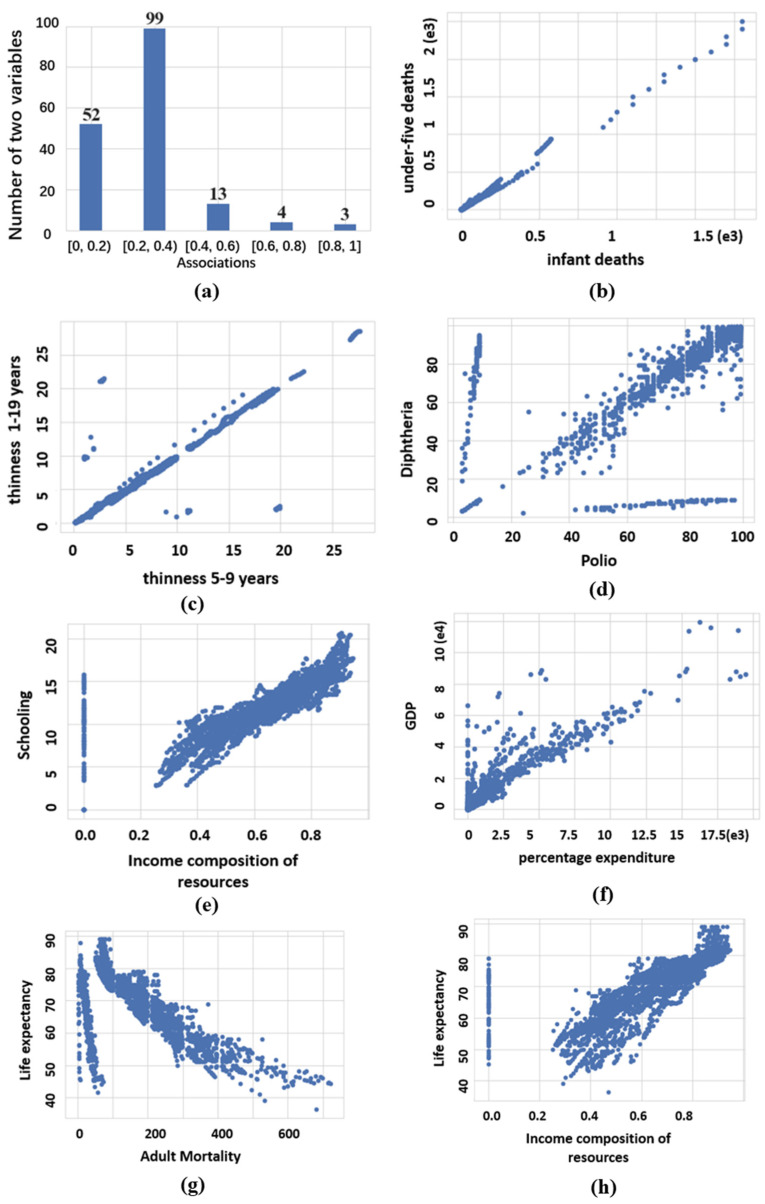
Exploring GHO dataset for bivariate associations with MIC.

**Table 1 sensors-22-02806-t001:** The color on a relationship and the variable on an axis in Figure 3.

Functions	*X*	*Y*	*Z*	Legend
1	Linear	Linear	Linear	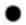
2	Linear	Linear × Cosine	Linear × Sine	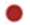
3	Linear	Polynomial	Sine + Linear	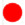
4	Linear	Piecewise Linear	Linear	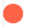
5	Linear	Cosine	Parabola	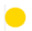
6	Linear	Exponential	Linear	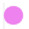
7	Linear	Sine	Logarithm	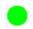
8	Linear	Exponential + Parabola	Cosine + Linear	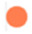
9	Linear	Sine	Cosine	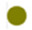
10	Polynomial	Cosine	Sine + Linear	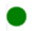
11	Linear	Polynomial	Polynomial	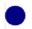
12	Linear	Power	Linear	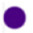

**Table 2 sensors-22-02806-t002:** The analysis of bias and variance of QOTIC, MTDIC, TEIC.

Method	Bias			Variance		
	Min	Mean	Max	Min	Mean	Max
QOTIC	0.011	0.096	0.143	0.0	0.001	0.003
MTDIC	0.017	0.11	0.164	0.0	0.002	0.005
TEIC	0.084	0.391	0.61	0.001	0.034	0.139

**Table 3 sensors-22-02806-t003:** Generality performance of MTDIC, QOTIC, and ESTIC.

Functions			*n* = 500			*n* = 1000	
		MTDIC	QOTIC	ESTIC	MTDIC	QOTIC	ESTIC
1	Linear	1.00	1.00	1.00	1.00	1.00	1.00
2	Exponential	1.00	1.00	1.00	1.00	1.00	1.00
3	Logarithmic	1.00	1.00	1.00	1.00	1.00	1.00
4	Quadratic	0.94	0.96	0.96	1.00	1.00	1.00
5	Cubic	0.96	0.97	0.97	0.97	0.98	0.98
6	Sinusoidal low freq.	0.87	0.92	0.92	0.94	0.95	0.95
7	Sinusoidal high freq.	0.58	0.59	0.59	0.74	0.75	0.75
8	Circle	0.49	0.50	0.50	0.55	0.56	0.56
9	Step function	1.00	1.00	1.00	1.00	1.00	1.00
10	Two lines	0.95	0.95	0.95	0.96	0.96	0.96
11	X line	0.47	0.51	0.51	0.53	0.55	0.55
12	X curve	0.48	0.54	0.54	0.53	0.56	0.56

**Table 4 sensors-22-02806-t004:** Time cost of MTDIC, QOTIC, and ESTIC.

	Functions	MTDIC	QOTIC	ESTIC
1	Linear	0.06	5.13	7.13
2	Exponential	0.07	5.13	7.13
3	Logarithmic	0.09	5.13	7.13
4	Quadratic	0.36	5.13	7.13
5	Cubic	0.19	5.13	7.13
6	Sinusoidal low freq.	0.41	5.13	7.13
7	Sinusoidal high freq.	0.40	5.13	7.13
8	Circle	0.34	5.13	7.13
9	Step function	0.08	5.13	7.13
10	Two lines	0.10	5.13	7.13
11	X line	0.39	5.13	7.13
12	X curve	0.37	5.13	7.13

**Table 5 sensors-22-02806-t005:** Trivariate associations by QOTIC and Bivariate associations by MIC.

Group	Trivariate Associations			QOTIC	Bivariate Associations		MIC
1	infant deaths	under-five deaths	life expectancy	0.942	infant deaths	life expectancy	0.31
					under-five deaths	life expectancy	0.342
					infant deaths	under-five deaths	0.958
2	thinness 1–19 years	thinness five to nine years	life expectancy	0.857	thinness 1–19 years	life expectancy	0.386
					thinness five to nine years	life expectancy	0.384
					thinness 1–19 years	thinness five to nine years	0.912
3	polio	diphtheria	life expectancy	0.777	polio	life expectancy	0.298
					diphtheria	life expectancy	0.295
					polio	diphtheria	0.801
4	percentage expenditure	GDP	life expectancy	0.685	percentage expenditure	life expectancy	0.31
					GDP	life expectancy	0.377
					percentage expenditure	GDP	0.714
5	income composition of resources	schooling	life expectancy	0.67	income composition of resources	life expectancy	0.614
					schooling	life expectancy	0.497
					income composition of resources	schooling	0.72
6	adult mortality	percentage expenditure	life expectancy	0.642	adult mortality	life expectancy	0.709
					percentage expenditure	life expectancy	0.31
					adult mortality	percentage expenditure	0.22
7	adult mortality	GDP	life expectancy	0.64	adult mortality	life expectancy	0.709
					GDP	life expectancy	0.377
					adult mortality	GDP	0.284

## Data Availability

The data presented in this study are openly available in [https://www.kaggle.com/datasets/kumarajarshi/life-expectancy-who, accessed on 28 February 2022].

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
