# Peer review of "Detecting Trivariate Associations in High-Dimensional Datasets"

_sensors, 2022, doi:10.3390/s22072806_

Round 1

Reviewer 1 Report

This paper proposes the quadratic optimized trivariate information coefficient (QOTIC) to measure the dependence among three variables equitably, for detecting the trivariate associations from complex relationships in high-dimensional datasets.

I have summarized by comments that can help to improve the paper:

The abstract of the paper should be more concert by describing the results of the proposed QOTIC compared with the state-of-the-art. It should discuss the takeaway message at the end of the abstract. 

In the introduction, it is recommended to express the motivation of this work in a more convincing way to understand its application in a real context. So that the general reader can get understood its applicability more easily. The improvement (as results (% improved?)) of the proposed method should be put in the introduction. 

The literature review should critically analyze the past work, not just summarize them. Please Identify the past work limitations and address how the proposed work solves these limitations.

Clear images of figures 1 and 2 are suggested. The text of the images is blurred. 

The conclusion should be self-contained. The definition of the abbreviation should not use before using it in the conclusion. 

The reference list should be updated and suggested to cite the recent work in this area. Only a few papers are up-to-date. 

Reviewer 2 Report

The paper is very interesting but has several critical issues:

The first criticality encountered is the definition of the field of action. The authors deal with the problem of the dependence between continuous variables, but speak of association while it generally refers to categorical variables (see: Shemis, E., & Mohammed, A. (2021). A comprehensive review on updating concept lattices and its application in updating association rules. Wiley Interdisciplinary Reviews: Data Mining and Knowledge Discovery, 11(2), e1401.; Montella, A., Mauriello, F., Pernetti, M., & Riccardi, M. R. (2021). Rule discovery to identify patterns contributing to overrepresentation and severity of run-off-the-road crashes. Accident Analysis & Prevention, 155, 106119.). you use an association to represent something like a field in a class. For numerical data we speak of absorbing.

The proposed methodology is unclear, the authors need to add a flow chart to explain the different steps.

Authors should enter the case study through variable drift and summary table introductions.

To compare the results the authors must use different performance measures not only the degree of implementation reported in table 2.

Reviewer 3 Report

In this manuscript, quadratic optimized trivariate information coefficient (QOTIC) is proposed to measure the dependence among three variables equitably, for detecting the trivariate associations from complex relationships in high-dimensional datasets. The principles of QOTIC are put forward. And a quadratic optimization procedure is presented to approach the correlation with high accuracy. QOTIC is more general and more equitable than existing methods on multivariable correlation for test functions, sample sizes, and noise levels. Finally, QOTIC finds out the trivariate associations with strong correlations from Global Health. Observatory dataset that is high-dimensional. Adequate revisions to the following points should be undertaken to justify the recommendation for publication.

  • Please rewrite an abstract section, justify an obtained result and contribution, improve a proposed method, etc.
  • This paper has more than spelling and grammatical errors. Please fix all of them.
  • The authors should clearly state the limitations of the proposed method in other applications.
  • Please write a contribution to this paper in the introduction section.
  • The related work section was very weak!! Please used 2022 and 2021 SCi indexed publications in Related Work. Why doesn’t the author use 2022 and 2021 publications?
  • All the figures have to be re-drawn with better quality.
  • Please change a “ Fundamental principles” title to “ 3. Proposed Method”, this section isn’t a fundamental principle, it is a proposed method.
  • Please change the “Conclusions” section to “Conclusions and Future WorkS” and write some future work.

Good luck

Round 2

Reviewer 1 Report

It seems the authors have done a sufficient level of improvement. As a suggestion, the authors should check the references again and try to incorporate the up-to-date ones. 
